# Vertical vs. Horizontal: How Strategic Alliance Type Influence Firm Performance?

**Baojun Yu [1], Hangjun Xu [2],\* and Feng Dong [3]**

1   Department of Management Science and Engineering, School of Management, Jilin University, Changchun 130022, China; yubaojun@jlu.edu.cn
2   Department of Marketing, McAfee School of Business, Union University, Jackson, TN 38305, USA
3   Department of Finance, School of Business, Siena College, Loudonville, NY 12211, USA; fdong@siena.edu
\*   Correspondence: jxu@uu.edu; Tel.: +1-5412855919

**Abstract:** Strategic alliances have become a key focus in the management and marketing literature. However, much of the previous research in this area has focused on the antecedents and accounting effects of strategic alliances. There is an opportunity to more closely examine how alliance types might influence the public equity markets. As a result, this study summarizes the literature for the theoretical foundation of strategic alliances to increase the understanding of the two main types of strategic alliances, that is industry scope (vertical vs. horizontal alliances) and size scope (asymmetric vs. symmetric alliances). Then, this study proposes a conceptual framework to examine the main and relative effects between different types of strategic alliances and firm performance. Using the Bloomberg Mergers and Acquisitions (M&A) database from 1 January 2010 to 1 January 2016, we find that vertical symmetric alliances gain more abnormal returns than others. Finally, implications and limitations are also discussed.

**Keywords:** asymmetric alliances; average abnormal return; horizontal alliances; strategic alliances; symmetric alliances; vertical alliances

## 1. Introduction

Over the last two decades, the merger and acquisition (M&A) and strategic alliance research have gained increasing popularity. Scholars have taken many different perspectives, such as the formation of strategic alliances [1–3], factors contributing to the success of alliances [4,5], and the outcome performance of alliances [6,7].

Although strategic alliances are often viewed as a key strategic resource and much research has found that the announcement of a strategic alliance can positively relate to firm abnormal returns [8–10], limited research has offered a complementary explanation for how different perspectives of alliance announcement types affect the stock market's evaluation, especially alliances with a firm's rivals or asymmetrical partners. Therefore, it is an imperative question for both scholars and practitioners: traditionally, alliances have been conceived of as ad hoc arrangements serving specific needs—is it worth engaging extensively in multiple simultaneous alliances to gain more competitive advantages?

To fill this research gap, this study summarizes the literature for the theoretical foundation of strategic alliances to increase the understanding of two main types of strategic alliances, that is industry scope (vertical vs. horizontal alliances) and size scope (asymmetric vs. symmetric alliances). Then, this study takes the case study findings of an alliance matrix [11] and extends their matrix to create a 2 (industry scope: vertical vs. horizontal alliances) * 2 (size scope: asymmetric vs. symmetric alliances) matrix to identify four different types of strategic alliances. We also propose that each of these alliance announcements can help the company gain positive firm abnormal returns and further examine

the relative relationship between strategic alliance type and firm alliance performance. Using the Bloomberg M&A database from 1 January 2010 to 1 January 2016, we find that vertical symmetric alliances gain more abnormal returns than others.

This study contributes to strategic alliance literature and practice. First, while interdependencies between business partners have been studied in the strategic alliance literature, the relationship between alliances of different types has not received due attention in prior studies. The most novel theoretical contribution for this study is that it not only shows the differential impacts of a firm's different types of strategic alliances (industry scope: vertical vs. horizontal alliances and size scope: asymmetric vs. symmetric alliances) on firm alliance performance, but it also discovers how a firm's vertical symmetric alliances gain more abnormal returns than others. Our fresh findings may extend strategic alliance and corporate governance literature by advancing our understanding of how good governance with business partners maximizes the value of the strategic alliance. Clearly, combining the different dimensions of alliance portfolios suggests a decent avenue for future research. Moreover, this study may offer essential suggestions to managers on where to allocate their precious resources and efforts, and when and how they collaborate with their organizational network partners to enhance competitive advantages.

## 2. Theoretical Foundations and Literature Review

### 2.1. Strategic Alliances and Firm Performance

Strategic alliances are contractual arrangements between two or more independent companies that carry out a project or operate in a specific business area by coordinating skills and resources jointly rather than either operating on their own or merging their operations [12]. From this definition, a strategic alliance must have two or more independent organizations join together to pursue mutual benefits, which will be greater than those from individual efforts.

This study stands on the resource-based view (RBV) perspective to examine how the type of strategic alliance influences firm performance, which is based on the role of resources, capabilities, and knowledge, in an attempt to further strategic objectives and create value [13–15]. As to business alliances, RBV researchers posit that *complementary* and *idiosyncratic* resources foster alliances to succeed [16]. Complementary resources are those that firms bring to an alliance that enable their alliance partners to fill out or complete their resource assortments [16–18]. The complementary resources could be tangible facilities or intangible resources, such as knowledge and connections [19,20]. Therefore, resource dependence theory (RDT), a sub-theory of RBV, states that inter-organizational relationships could also help an organization to reduce environment uncertainty and gain mutual benefits [21,22], which is used for explaining why firms engage in long-term relationships with other firms. Idiosyncratic resources are defined as those that (1) are developed during the life of the alliance, (2) are unique to the alliance, and (3) facilitate the combining of the distinct lower-order resources contributed by the partner firms (and, hence, are higher-order resources) [23]. From an RBV perspective, idiosyncratic resources, since they are unique to the alliance and are constantly evolving, help alliances maintain the durability and inimitability of their resource advantage [16,24].

Because of the growing importance of strategic alliances and the fact that as many as 70% of all alliances are judged unsuccessful [25], it is not surprising that previous research focused on alliance outcomes. Alliance research has evolved in three streams: (1) The alliance literature; (2) Studies of stock market returns following alliance announcements; (3) Social network theory applications [26]. Most early studies on the second stream of alliance research only identified one dimension of alliance announcements that influence alliance outcomes. Chan and his colleagues (1997) investigated share price responses to the formation of 345 strategic alliances spanning 1983–1992. They found that strategic alliance announcements in the US had significant announcement day abnormal returns of 0.64% [7]. Very consistent with this empirical finding, Gleason, Mathur, and Wiggins (2003) reported significant announcement day abnormal returns of 0.66% for strategic alliances in the US Financial

Services industry [27]. Some other empirical research focused on particular strategic alliances to test the relationship between strategic alliance announcements and firm performance. For example, Chan and his colleagues (1999) also examined 345 technical and marketing alliances and concluded that the overall average abnormal return is around 0.64%, while the high-technology firms can gain more benefit, amounting to 1.12%, and the overall average abnormal return benefit by technical horizontal alliances and marketing non-horizontal alliances can be even higher, up to 3.5% and 1.45%, respectively [28]. Swaminathan and Moorman (2009) only picked up 230 marketing alliances and found that marketing alliance announcements can create an average abnormal return value of 1.4% [10]. These research streams enhance our understanding of alliances and firm performance but fall short of fully accounting for how different perspectives of alliance announcement types affect the stock market's evaluation. Therefore, it is still imperative to answer the question: how does a firm develop and choose strategic alliance partners to improve and maintain their competitive advantages?

### 2.2. Strategic Alliance Types

Yasuda and Iijima (2005) used symmetric and asymmetric alliances as the first dimension to direct the nature of resources. In symmetric alliances, the same kinds of resources are exchanged, while in asymmetric alliances, different kinds of resources are exchanged. The second dimension they used was horizontal and vertical alliances—a horizontal alliance is one in which the partners belong to the same industry, while a vertical alliance is one in which the partners are from different industries [11].

This study took the case study findings of the alliance matrix [11] and extended their matrix to examine the strategic alliance's impact on firm alliance performance. Different from Yasuda and Iijima (2005)'s alliance matrix, we categorized strategic alliances according to the same dimensions but different perspectives. In our matrix, we agree with the categorization of horizontal and vertical strategic alliances as the industry scope of strategic alliances, which distinguishes whether the alliance partner is the rival competitor for market share in the same industry or different industries [11,18]. While, following some previous empirical research related to asymmetric and symmetric alliances, we involved disparately sized firms to define asymmetric and symmetric alliances [29]. In the following section, we aim to clarify these two dimensions.

***Industry scope of strategic alliance.*** *Vertical strategic alliances* describe the collaboration between a company and its upstream and downstream partners in the supply chain, which means a partnership between its suppliers and distributors [10,30,31], especially one in which suppliers get involved in product design and distribution decisions. *Horizontal strategic alliances* are formed by firms that are active in the same business area. Such partners in the alliance used to work together to improve market power compared to other competitors. The similarity of these two types of alliances lies in the following several perspectives: (1) Partners are confident with the advice and recommendations suggested by trusted business relationships [11,30], so such organizational relationships may reduce uncertainty and costs in business transactions; (2) Information sharing with business partners can foster knowledge spillover and production [30]; (3) By involving relationship participators to co-work through problems, joint problem-solving arrangements may replace the simple exit-or-stay response of the market players [30]. These two types of alliances also have distinct features: (1) Key partners—vertical alliances mainly focus on their suppliers and customers, while horizontal alliances focus on their main competitors; (2) Key activities—vertical alliances try to co-operate with their partners by sharing raw materials, production, and distribution to maximize profits, while in the case of horizontal alliances, by reducing vicious competition, common potential market opportunity is pursued to create synergies among competitors; (3) Key perspectives—the high quality of vertical alliances can offer complementary information and knowledge to achieve organizational cooperation. Although the information and knowledge in the same industry are similar, firms in many knowledge-intensive industries force themselves to seek strategic alliances with major competitors with whom they have joint interests in some markets and/or product fields [31]. Table 1 summarizes the empirical results of the literature on the industry scope of strategic alliances.

**Table 1.** A review of the literature on the industry scope of strategic alliances.

| Authors | Focus | Contexts and Data Source | Method | Main Findings |
|---|---|---|---|---|
| Su Han Chan et al. (1999) | Horizontal alliances or non-horizontal technical and marketing alliance | 345 announcements (LexisNexis database and Dow Jones News Retrieval Service database: 1983–1992) | Event study and regression | Average abnormal return (0.64%); high-technology firms (1.12%); technical horizontal alliances (3.5%) and marketing non-horizontal alliances (1.45%) |
| Rindfleisch (2000) | Horizontal and vertical research & development alliances | 106 US firms (Federal Register) | Hierarchical regression analyses | Participants in vertical alliances display higher levels of organizational trust than in horizontal alliances |
| Luo et al. (2007) | Horizontal alliances | 228 US firms (Standard and Poor's Compustat database) | Hierarchical regression analyses | The intensity of a firm's alliances with its competitors has a curvilinear (inverted U-shaped) influence on return on equity |
| Oxley et al. (2009) | Horizontal R&D-related alliances | 241 alliances (Securities Data Company Database) | Event study and regression | Horizontal alliances positively related to cumulative abnormal return |
| Swaminathan and Moorman (2009) | Marketing alliances | 230 announcements (Securities Data Company Joint Ventures and Strategic Alliances database) | Event study | Marketing alliance announcements create value (1.4%). Network efficiency and network density have the strongest positive impact when they are moderate |
| Belderbos et al. (2012) | Horizontal and vertical technology alliances | Panel set innovation firms in Netherlands (1996–2004) | Multivariate probit model | Vertical alliances exhibit a higher degree of persistence than horizontal alliances |

*Size scope of strategic alliance.* Kalaignanam, Shankar, and Varadarajan (2007) considered asymmetric alliances as those alliances in which the ratio of the larger firm's assets to that of the smaller firm is greater than five [29]. Following their work, we also consider the symmetric alliances as those alliances in which the ratio of the firm's assets to that of the other firm is close to or less than five. Previous research suggests that the size of a potential partner is an important criterion in partner selection. Firstly, let us take a look at the motivation of symmetric alliances. A possible reason is that both firms would place the same importance on the alliance and bargaining power would be almost equal. Similarly, Williams and Lilley (1993) argued strongly that an alliance may have the best chance of long-term success when both partners are comparable in sophistication and size [32]. On the other hand, earlier research analyzed the asymmetric distribution of common and private benefits in alliances and underscored the incentives that such benefits provide for continued collaboration. The major reason for asymmetric alliances is to access complementary resources from each other [33]. For example, small biotech firms often form alliances with large pharmaceutical firms with the purpose of utilizing the latter's expertise in the US Food and Drug Administration agency's approval process and in market coverage. Table 2 summarizes the empirical results of the literature on the size scope of strategic alliances.

**Table 2.** A review of the literature on the size scope of strategic alliances.

| Authors | Focus | Contexts and Data Source | Method | Main Findings |
|---|---|---|---|---|
| Chen and Hambrick (1995) | Asymmetric alliance | 28 US major airlines (1985–1986) | T-tests and Multivariate analysis of variance | The small airlines had a greater propensity for action and faster speed for executing action than large firms |
| Stuart (2000) | Asymmetric high-technology alliance | 1600 dyadic alliances | Poisson regression | Technology alliance with large partners improved baseline innovation and growth rates, while the small firm had an immaterial effect on performance |
| Kalaignanam et al. (2007) | Asymmetric new product development (NPD) alliances | 167 alliances between larger and smaller firms (SDC and LexisNexis) | Event study | Both the partners experienced significant short-term financial gains, but there were considerable asymmetries between the larger and smaller firms with regard to the effects of alliance, partner, and firm characteristics on the gains of the partner firms |

## 3. Conceptual Framework and Hypothesis Development

Looking deeper into the relationship between strategic alliance type and performance, four different types of the strategic alliance were identified. We propose that each of these alliance announcements can help the company gain positive firm abnormal returns. Then, we argue that there is a relative proportion of these four different types of strategic alliances.

### 3.1. Vertical Strategic Alliance and Firm Performance

We defined vertical asymmetric alliances as a larger firm cooperating with a smaller firm in a different industry, while we defined vertical symmetric alliances as a larger or smaller firm cooperating with a close-sized firm in a different industry. According to resource dependence theory (RDT), interdependence is a phenomenon that "exists whenever one actor does not entirely control all of the conditions necessary for the achievement of an action or for obtaining the outcome desired from the action" [34]. As RDT suggests, organizations form inter-organizational relationships with other organizations as a governance mechanism to reduce uncertainty and manage dependence [35]. Therefore, we argue that both vertical asymmetric alliances and vertical symmetric alliances can help the company gain positive firm abnormal returns for the following three reasons: (1) Business partners are confident with the advice and recommendations suggested by trusted relationships, so such trustable organizational relationships with upstream and downstream partners can simplify transaction procedures and reduce uncertainty and transaction costs [36–38]; (2) The very turbulent dynamic markets in the modern society make information sharing with the upstream and downstream firms more necessary and urgent—a sharing practice that helps the firms keep updated with the rapid market and technological changes, so that the firms are able to swiftly adjust their innovation strategy [39]; (3) When the firms listen to their suppliers and customers through the joint problem-solving mechanism, their respect to the partners will be well received, which will help the firms maintain good and firm relationships with the partners [40,41]. So:

**H1a:** *Vertical asymmetric alliances gain positive firm abnormal returns.*

**H1b:** *Vertical symmetric alliances gain positive firm abnormal returns.*

### 3.2. Relative Effects of the Two Types of Vertical Strategic Alliances

As we listed above, both of the vertical alliances aim at intensifying and improving these relationships and enlarging the company's network in order to be able to offer lower prices. As for the larger firms, alliances with symmetric partners in their upstream and downstream network will help

them access more resources to maintain a competitive advantage [38]. In addition, vertical symmetric companies can typically offer greater "staying power", being able to commit a greater volume of resources over a longer time horizon [32]. Therefore, they argued that joint ventures have the best chance of long-term success when both partners are comparable in sophistication and size [32]. Stuart (2000) also took the high-technology industry as the research context and concluded that technology alliances with symmetric and larger partners improved baseline innovation and growth rates, while alliances with smaller firms had an immaterial effect on performance. Moreover, significant size differences between vertical alliances led to other problems [33]. One concern is the possibility of the domination of one firm over the other. The smaller firm may share its innovative technology with a larger firm offering finance, marketing, distribution resources, etc. until the larger firm learns and executes the technology. This vertical asymmetric alliance could not have a longer duration, because the small firm could be a burden for the larger firm. Therefore, we made the following proposition:

**H2:** *Vertical symmetric alliances gain more abnormal returns than vertical asymmetric alliances.*

### 3.3. Horizontal Strategic Alliance and Firm Performance

We also defined horizontal asymmetric alliances as a larger firm cooperating with a smaller firm in the same industry, while we defined horizontal symmetric alliances as a larger or smaller firm cooperating with a close-sized firm in the same industry. We argue such horizontal strategic alliances with competitors in the same industry might also improve abnormal returns in the three processes aforementioned, but for different reasons: (1) The literature on coopetition (collaboration while in competition) suggests that intense competition is mitigated if firms are tied to each other [7,42,43]. This is because, with trust and long-term relationships, firms likely believe that the other will not engage in certain practices, such as charging overly low prices and behaving unethically [44]. When facing less vicious competition, firms are more likely to spend their limited resources on new product development to keep pace with the dynamic market changes; (2) Although firms in the same industry face the same market environment, the knowledge and resources owned by each firm are different. Through sharing and integrating their product development knowledge and skills, the firms might use their knowledge and resources more efficiently, so that they can improve the accuracy and efficiency of marketing sense capability to jointly develop and launch new products faster [31,45,46]; (3) Collaboration with competitors can help firms acquire mutual benefits through joint problem solving. In particular, they may gain more bargaining power when negotiating with suppliers [47], so that the firms may access constrained supplies or offer products and services at relatively lower and more competitive price. Therefore:

**H3a:** *Horizontal asymmetric alliances gain positive firm abnormal returns.*

**H3b:** *Horizontal symmetric alliances gain positive firm abnormal returns.*

### 3.4. Relative Effects of the Two Types of Horizontal Strategic Alliance

Previous coopetition research has suggested that firms can gain positive financial performance by blending competition with cooperation [48–50]. Luo, Rindfleisch, and Tse (2007) also found that these alliances could enhance profitability by mixing the benefits of both competition (e.g., efficient resource allocation) and cooperation (e.g., enhanced information flow) [49]. When we consider the symmetry or asymmetry of the horizontal alliances, it is well known that greater interdependence and uncertainty in an alliance may increase coordination costs [51]. The larger firm has a stronger power to gain control of the alliance to reduce the coordination costs, especially in the same industry.

Moreover, large and strong firms may be willing to form strategic alliances with small and weak firms in the same industry because they hold the power to appropriate relational rent, and alliances with inferior firms in a sub-network enhance the bargaining power of the superior firms in the entire network [52]. Gomes-Casseres (1997) also noted that larger firms have been traditionally dominant

players in the Information Technology and Pharmaceutical industries. The advent of new technologies presents unique opportunities for smaller entrepreneurial firms in the same industry to pursue targeted innovation [53]. Therefore, such horizontal asymmetric partners can share their information or knowledge efficiently to speed their new product to market [39]. Therefore, we argue that an alliance with asymmetric partners gains more benefits for shareholders.

**H4:** *Horizontal asymmetric alliances gain more abnormal returns than horizontal symmetric alliances.*

## 4. Methodology

### 4.1. Sample and Data Sources

We used the Bloomberg M&A database to collect data. We took the following steps to generate our sample of alliances from this database: We first included all of the five deal types in this database, that is M&A, investment, joint venture, spin-off, and buyback. We also included all of the five statuses in this database, that is proposed, pending, completed, withdrawn, and terminated. We selected the time period from 1 January 2010 to 1 January 2016. Both of the target and acquirer firms are publicly listed on US, the complete data of which were available from the other databases. Both of the target and acquirer firms are public and should include the Current Standard Industrial Classification (SIC) Code. So far, we obtained 2083 alliance announcements in total. After looking at the data, we found that in 235 alliance announcements in the buy deal type, the target and the acquirer firm were the same. We also found that there were 1539 M&As. Therefore, we decided to delete these two parts, and finally we obtained our sample size as the 305 alliance announcements of joint ventures.

### 4.2. Variable Measurements

*Vertical* vs. *horizontal alliances*: We used the target and acquirer Current Standard Industrial Classification (SIC) Code to classify the vertical or horizontal alliances. By using the first two-digit SIC code instead of the four-digit SIC code of the target and acquirer [54], we considered the target and acquirer located at the same code range as belonging to the same industry. For example, the North American Industry Classification System (NAICS) listed the first two-digit SIC code range 01–09 as the Agriculture, Forestry, and Fishing industry, 10–14 as the Mining industry, 15–17 as the Construction industry, 20–39 as the Manufacturing industry, 40–49 as the Transportation, Communications, Electric, Gas, and Sanitary industry, 50–51 as the Wholesale Trade industry, 52–59 as the Retail Trade industry, 60–67 as the Finance, Insurance, and Real Estate industry, 70–89 as the Services industry, 91–99 as the Public Administration industry.

*Symmetric* vs. *asymmetric alliances:* Following Kalaignanam, Shankar, and Varadarajan (2007), we considered asymmetric alliances as those alliances in which the ratio of the larger firm's assets to that of the smaller firm was greater than five [29].

*Control variables*: Firm age, operationalized as the time elapsed from the date of the founding of the firm to the date of the alliance announcement [29]; firm sales, defined as the dollar amount of actual billings for regular sales completed during the period, reduced by cash and trade discounts (Dutta, Narasimhan, and Rajiv, 1999); alliance experience (1 = at least one alliance in the past 10 years, 0 = this is the first alliance in the past 10 years) [55]; log (size), which is the log value of the firm's market capitalization; Return on Assets (ROA), which is the return on the total assets ratio; and the price to book ratio of the company. All control variables were estimated using firm level data one year prior to the announcement date.

### 4.3. Data Analysis

We modeled the impact of predictor variables on abnormal stock returns accruing to the firm as a consequence of an alliance. Firstly, we developed each alliance model as follows: $R_{ijt} = a_i + \beta_i R_{mjt} + \epsilon_{ijt}$ (where $R_{ijt}$ is the firm i's stock return in its alliances j on that day t; $R_{mjt}$ is the market index return

in the alliances j on that day t; $\epsilon_{ijt}$ is the random-error term), then we calculated daily abnormal returns as follows: $AR_{ijt} = R_{ijt} - (a_i + \beta_i R_{mjt})$ (where $AR_{ijt}$ is the daily abnormal returns for firm i's stock return in its alliances j on that day t). Following previous research, we defined the event window as a period of five trading days centered on the event day (day 0) [56]. Finally, the cumulative abnormal returns (CARs) from day −5 to day +5 were calculated.

Table 3 shows the abnormal returns calculated across 305 alliance announcements and the aggregated cumulative abnormal returns over five trading days. The cumulative abnormal return on event day 0 was not significant. We found that the cumulative abnormal return for vertical alliances from day −5 to day −1 (2.6%) was significantly bigger than horizontal alliances (−1.0%) and we found that same pattern for the abnormal return on event days −5 and +5 (4.4% > −2.3%). The cumulative abnormal return for symmetric alliances from day −5 to day −1 (2.7%) was significantly bigger than asymmetric alliances (−1.4%) and we found that same pattern for the abnormal return on event days −5 and +5 (2.5% > −1.0%).

**Table 3.** Cumulative abnormal returns across various event windows.

|  |  | Event Window | | | |
|---|---|---|---|---|---|
|  |  | (−5, −1) | (0) | (+1, +5) | (−5, 5) |
| All alliances | Parameter | 0.00639 | 0.01062 | 0.00164 | 0.00803 |
|  | t Value | 0.31 | 1.30 | 0.12 | 0.24 |
| Horizontal alliances | Parameter | −0.01008 | 0.01774 | −0.01242 | −0.0225 |
|  | t Value | −0.28 | 1.35 | −0.52 | −0.39 |
| Vertical alliances | Parameter | 0.02553 | 0.00660 | 0.01836 | 0.04389 |
|  | t Value | 1.65 * | 0.67 | 1.41 | 2.25 ** |
| Symmetric alliances | Parameter | 0.02687 | 0.01754 | −0.00202 | 0.02485 |
|  | t Value | 2.46 ** | 1.60 | −0.26 | 1.93 * |
| Asymmetric alliances | Parameter | −0.01408 | 0.00799 | 0.00626 | −0.00782 |
|  | t Value | −0.39 | 0.5281 | 0.25 | −0.13 |
| Horizontal symmetric alliances | Parameter | 0.02139 | 0.02187 | −0.01211 | 0.00928 |
|  | t Value | 1.35 | 1.25 | −1.10 | 0.51 |
| Horizontal asymmetric alliances | Parameter | −0.05914 | 0.01154 | −0.01629 | −0.07543 |
|  | t Value | −0.85 | 0.52 | −0.35 | −0.66 |
| Vertical symmetric alliances | Parameter | 0.04203 | −0.00343 | 0.01240 | 0.05443 |
|  | t Value | 1.74 * | −0.21 | 0.64 | 1.79 * |
| Vertical asymmetric alliances | Parameter | 0.03465 | 0.00572 | 0.01263 | 0.04727 |
|  | t Value | 1.51 | 0.40 | 0.66 | 1.64 * |

Notes: *** $p < 0.01$, ** $p < 0.05$, * $p < 0.1$.

H1a and H1b attempt to examine how both vertical asymmetric and vertical symmetric alliances may gain positive firm abnormal returns. We found that the cumulative abnormal return for vertical asymmetric alliances from day −5 to day −1 was 4.2%, significantly at the 0.1 level, which supports H1b. However, we did not find statistically significant evidence to support our H1a. H2 investigates how the vertical symmetric alliances can gain more abnormal returns than vertical asymmetric alliances. From the above empirical results, we found that H2 can be supported. We also found that same pattern for the abnormal return on event days −5 and +5 (5.4% > 4.7%). Surprisingly, we did not find any other evidence to support the following three hypotheses related to the horizontal symmetric and horizontal asymmetric alliances.

We also conducted a multivariate regression analysis by regression firm 10-day (−5, 5) CARs around announcement day on dummy variables indicating vertical alliances (Vertical Dummy = 1), symmetric alliances (Symmetric Dummy = 1), and vertical symmetric deals (Vertical * Symmetry = 1), controlling other firm level characteristics. We set up dummy variables based on the results, as presented

in Table 3, that vertical and symmetric deals yield significantly higher CARs than others. The regression results are reported in Table 4.

**Table 4.** Multivariable analysis of company performance on vertical and symmetric alliances.

|  | (−5, 5) CAR (%) | | | |
|---|---|---|---|---|
|  | **[1]** | **[2]** | **[3]** | **[4]** |
| Vertical Dummy | 10.13784 |  | 10.55051 * |  |
|  | (1.61) |  | (1.68) |  |
| Symmetric Dummy |  | 17.57139 ** | 17.76671 *** |  |
|  |  | (2.55) | (2.58) |  |
| Vertical*Symmetric |  |  |  | 23.88053 ** |
|  |  |  |  | (2.45) |
| Log (Size) | −0.31204 | 0.03536 | 0.14447 | 0.15202 |
|  | (−0.26) | (0.03) | (0.12) | (0.12) |
| Return of Assets (ROA) | 0.06408 | 0.00637 | 0.01121 | 0.0298 |
|  | (0.58) | (0.06) | (0.10) | (0.27) |
| Price to Book | 0.02776 | 0.01545 | −0.00747 | −0.01512 |
|  | (0.07) | (0.04) | (−0.02) | (−0.04) |
| Intercept | −0.46165 | −3.97414 | −8.87329 | −1.9429 |
|  | (−0.06) | (−0.47) | (−1.00) | (−0.24) |
| Adjust R-Square | 0.0036 | 0.0094 | 0.0154 | 0.0078 |
| Number of Observations | 302 | 299 | 299 | 299 |

Notes: *** $p < 0.01$, ** $p < 0.05$, * $p < 0.1$.

The results shown in Table 4 further confirmed our hypothesis that vertical (10.55051, t = 1.68, as in regression [3]) and symmetric (17.76671, t = 2.58, as in regression [3]) alliances lead to higher firm abnormal returns around announcement days. Furthermore, the interactive variable, Vertical * Symmetry, shows a significant positive relation between vertical symmetric alliances and firm performance (CARs). Table 4 offers strong evidence that industry scope (vertical vs. horizontal alliances) and size scope (asymmetric vs. symmetric alliances) play essential roles in strategic alliance outcomes.

To further test the differential impacts of the different types of strategic alliances on a firm's long-term performance after the announcement, we regressed a firm's operating efficiency, which is estimated as the firm's operating income/loss over total firm assets, in the first, second, and third year after the announcement year, on firm level control variables. The results are presented in Table 5.

The long-term firm efficiency analysis, along with our short-term firm performance analysis results, show that vertical and symmetric alliances improve a company's performance significantly and consistently. The positive relation exists for at least three years. To exam the sensitivity of our results, we also used return on invested capital (ROIC), which measures the percentage return from their invested capital, as an alternative firm efficiency measurement, and the results are highly consistent, as shown by the results in Table 5.

**Table 5.** Multivariable analysis of company long-term efficiency on vertical and symmetric alliances.

| | Operating Income | | | | | | | | |
| | One Year after Announcement | | | Two Years after Announcement | | | Three Years after Announcement | | |
|---|---|---|---|---|---|---|---|---|---|
| Vertical Dummy | 0.098403 * | | 0.101069 ** | 0.220634 * | | 0.221164 * | 0.254759 ** | | 0.253414 ** |
| | (1.93) | | (2.02) | (1.88) | | (1.88) | (2.53) | | (2.51) |
| Symmetric Dummy | | 0.179181 *** | 0.175606 *** | | 0.219822 * | 0.217770 * | | 0.072113 | 0.066710 |
| | | (3.41) | (3.37) | | (1.79) | (1.79) | | (0.68) | (0.64) |
| Log (Size) | −0.018687 * | −0.018990 ** | −0.016243 * | −0.002504 | −0.006126 | 0.002564 | 0.006194 | −0.000111 | 0.008259 |
| | (−1.92) | (−2.00) | (−1.71) | (−0.11) | (−0.27) | (0.11) | (0.32) | (−0.01) | (0.43) |
| Return of Assets (ROA) | 0.000593 | 0.000141 | 0.000125 | 0.000637 | −0.000125 | −0.000069 | 0.004817 *** | 0.004590 ** | 0.004619 ** |
| | (0.67) | (0.16) | (0.14) | (0.32) | (−0.06) | (−0.03) | (2.65) | (2.44) | (2.50) |
| Price to Book | −0.003676 | −0.004102 | −0.004133 | −0.000771 | −0.001319 | −0.001578 | 0.015957 | 0.016348 | 0.015370 |
| | (−1.31) | (−1.49) | (−1.51) | (−0.13) | (−0.22) | (−0.27) | (0.98) | (0.99) | (0.94) |
| Intercept | −0.002840 | −0.022692 | −0.078594 | −0.254227 | −0.220694 | −0.360316 ** | −0.352838 ** | −0.239521 * | −0.387813 *** |
| | (−0.04) | (−0.35) | (−1.13) | (−1.59) | (−1.44) | (−2.12) | (−2.58) | (−1.74) | (−2.63) |
| Adjust R-Square | 0.0272 | 0.0614 | 0.0743 | 0.001 | −0.0027 | 0.0113 | 0.0628 | 0.0236 | 0.0589 |
| Number of Observations | 228 | 226 | 226 | 185 | 184 | 184 | 146 | 146 | 146 |

Notes: *** $p < 0.01$, ** $p < 0.05$, * $p < 0.1$.

## 5. Conclusion

### 5.1. Conclusions

This study summarized the literature for the theoretical foundation and definition of strategic alliances to increase the understanding of the types of strategic alliances. Then, by extending Yasuda and Iijima (2005)'s alliance matrix, we identified four different types of strategic alliances [11]. We also proposed that each of these alliance announcements helped the company gain positive firm abnormal returns and further examined the relative relationship between strategic alliance type and firm alliance performance. Using the Bloomberg M&A database from 1 January 2010 to 1 January 2016, we found that vertical symmetric alliances gain more abnormal returns than others.

### 5.2. Theoretical Implications

This study contributes to the literature in several notable ways. First, this study extends the resource dependence theory (RDT) [21,22] and strategic alliance literature [12,25,26] by empirically testing the differential impacts of the firm's different types of strategic alliances on firm alliance performance. We hope that the idea of integrating different types of strategic alliances may pave the way for future empirical studies to maintain their inter-organizational relationships to gain competitive advantages.

Moreover, this study enriches the research on the marketing–finance interface by investigating how firms' vertical symmetric alliances gain more abnormal returns than others. Such findings respond to Yasuda and Iijima 's (2005) [11] call for more research on their matrix in bridging firms' inter-organizational resources and firm alliance performance.

Third, according to previous corporate governance literature [57], weak governance firms have lower equity returns, worse operating performance, and lower firm value. The current findings in this study may enrich corporate governance literature by creating and maintaining good governance to maximize the value from their vertical symmetric strategic alliances.

### 5.3. Managerial Implications

This study shows that strategic alliances can lead to a number of positive outcomes for firms. Therefore, managers should develop an open mindset to connect the external environment with internal organizational capability development. Our empirical findings show that managers should pay more attention to the formation of vertical symmetric alliances, which can create more abnormal returns.

The coopetition between Sony and Samsung is a good example of addressing major technological challenges, creating benefits for partnering firms, and advancing technological alliances [58,59]. For many years, Samsung Electronics' key mission was to beat Sony Corporation as the world's top electronics maker, and both Sony and Samsung competed vigorously in many electronic product–market segments. Despite their fierce rivalry, the two firms established a joint venture (S-LCD Corporation) in April 2004 to develop and produce seventh generation liquid crystal display (LCD) panels for flat-screen televisions. Samsung contributed its technological strengths regarding the LCD technology while Sony contributed its technological strengths and brand recognition regarding televisions [60].

### 5.4. Limitations and Future Research

This study has several limitations. First, by choosing Yasuda and Iijima (2005)'s alliance matrix as the foundation of this study, we only focused on these two main types of strategic alliances as the search terms in the present study and some other matrices may be inevitably excluded. In the future, we want to create a broader and more complex matrix to help us to identify important contributions using other types of strategic alliances. Moreover, we obtained data from the Bloomberg M&A database through 1 January 2010 to 1 January 2016 and obtained only 305 alliance announcements of joint ventures. The limited sample might have reduced the statistical power necessary to generate more significant findings. Further research could test our hypotheses using larger samples. Third, this study

mainly focused on exploring the relationship between different types of alliances and firm performance. More and more empirical tests may also provide insights on the firm-level antecedents of different types of alliances. Finally, it was only focused on the bright side of strategic alliances, while there are many cases of failure regarding strategic alliances and some strategic alliances have a very short duration. In the future, there should be an attempt to find some boundary conditions and empirical tests interested in the marketing or management discipline.

**Author Contributions:** Conceptualization, B.Y. and H.X.; methodology, F.D.; writing—original draft preparation, H.X.; writing—review and editing, H.X.

**Funding:** This research was funded by The Frontier and Innovative Project of Philosophy, Social Science and Interdisciplinary in Jilin University, grant number 2019QY017.

**Acknowledgments:** The authors thank the Editor and three anonymous reviewers for their insightful comments and guidance.

**Conflicts of Interest:** The authors declare no conflict of interest.

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
