# Peer review of "Vertical vs. Horizontal: How Strategic Alliance Type Influence Firm Performance?"

_sustainability, doi:10.3390/su11236594_

Round 1

Reviewer 1 Report

This research applies event study to explore the influence of strategic alliance type to firm performance. However, the authors provide only one testing result of "Cumulative abnormal returns across various event windows" to support their hypothesis, and the evidence is apparently weak. The following testing works are strongly recommended.

Distribution of Sample Firms in Announcement Year Descriptive Statistics of Sample Characteristics Regressions of CAR With Respect to Firm Characteristics Changes in Performance Indicators after the event day

Author Response

Sustainability-604157: Response to Comments of Reviewer 1

We provide responses to each of your specific comments (your comments are excerpted in bold fond, followed by our responses in normal type).

R1-1 This research applies event study to explore the influence of strategic alliance type to firm performance. However, the authors provide only one testing result of "Cumulative abnormal returns across various event windows" to support their hypothesis, and the evidence is apparently weak. The following testing works are strongly recommended: Distribution of Sample Firms in Announcement Year Descriptive Statistics of Sample Characteristics Regressions of CAR With Respect to Firm Characteristics Changes in Performance Indicators after the event day.

Response:

Thank you for your helpful suggestion. Following your suggestion, we conducted a multivariate regression analysis by regression firm 10-day (-5, 5) CARs around announcement day on dummy variables and reported our result on the new table 4. For the sake of brevity, we do not cite the text here. Please see the content from line 275-289 for details.

Thank you very much for your time and efforts in helping us improve the paper. Your detailed and constructive comments encouraged us to think more deeply and broadly about our empirical section. We hope you find that the current manuscript is much improved in its empirical part.

Reviewer 2 Report

The paper is very interesting and actual nowadays when companies always want to improve their efficiency and productivity. Paper is mainly descriptive, but in that case is not a problem, because authors synthesized related literature and after prepared their own matrix, which is a new one. In line 257  there is a spelling mistake: "... both vertical asymmetry and vertical asymmetry..." my offer based on the research work would be the following: ...both vertical asymmetry and vertical symmetry...

Discussion part is correct and easy to understand.

It is not necessary to write from line 299-328 in that form.

Author Response

Sustainability-604157: Response to Comments of Reviewer 2

We provide responses to each of your specific comments (your comments are excerpted in bold fond, followed by our responses in normal type).

R2-1 The paper is very interesting and actual nowadays when companies always want to improve their efficiency and productivity. Paper is mainly descriptive, but in that case is not a problem, because authors synthesized related literature and after prepared their own matrix, which is a new one. In line 257  there is a spelling mistake: "... both vertical asymmetry and vertical asymmetry..." my offer based on the research work would be the following: ...both vertical asymmetry and vertical symmetry...

Discussion part is correct and easy to understand.

It is not necessary to write from line 299-328 in that form.

Response:

We appreciate your constructive and helpful comments very much. Moreover, thank you for pointing out the typographical error in the previous manuscript. We have corrected it in the revision. Again, we have invited a professional copy editor to proofread the manuscript to ensure the quality of the writing. By the way, due to the format of this Journal, editor adds the content from line 299-328 in our original manuscript. We believe that our manuscript has benefited immensely from your comments, and for that we are extremely grateful.

Reviewer 3 Report

attached

Author Response

Sustainability-604157: Response to Comments of Reviewer 3

We provide responses to each of your specific comments (your comments are excerpted in bold fond, followed by our responses in normal type).

R3-1 You should clarify the contributions of the paper which are not elaborated well in the current paper. You can talk about the following contributions: What insights can you provide based on your finding? Do they push forward our understanding? What should we do with your research? Do you have any suggestions to improve the current regulation or practice? Adding the above discussion and extend your literature review may help you make more contributions and position your contributions better.

Response:

Thank you for the constructive comment. In this revision, we made the major changes in our introduction to address your concern. For the sake of brevity, we do not cite the text here. Please see the content from line 42-50 for details.

R3-2 You should study and rationalize the use of firm size measures in the literature since frim size is the key variable in this area. See Dang et al. 2018. Measuring Firm Size in Empirical Corporate Finance. Journal of Banking & Finance, 86:159-176. After all it is the most significant variable in most studies alike. You need to discuss and justify your firm size measure.

Response:

Thanks for your valuable comment. We do acknowledge that the firm size measure is the key variable in this area. Actually, one of the type of strategic alliance is size scope (asymmetric vs. symmetric alliances). Following the previous literature (Kalaignanam, Shankar and Varadarajan, 2007), we consider asymmetric alliances as those alliances in which the ratio of the larger firm’s assets to that of the smaller firm is greater than five.

R3-3 My main suggestion is that you should tell a richer story and link to more literature by discussing more relevant channels. One main channel is corporate governance. The abnormal returns of different alliances may be driven by agency problem, for instance. You should consider, for example, market competition as a governance mechanism: Giroud, X., and H., Mueller, 2011, Corporate governance, product market competition, and equity prices. Journal of Finance 66, 563-600. The interactions between the executives, such as mutual monitoring among the executives: Li, Z.F., 2014, Mutual monitoring and corporate governance, Journal of Banking & Finance, 45, 255-269; Li, Z.F., 2018, Mutual monitoring and agency problem. 
https://www.researchgate.net/publication/272305464_Mutual_Monitoring_and_Agency_Problems; and external interactions between CEOs in the industry tournament: Coles et al. 2018, Industry Tournament Incentives, Review of Financial Studies, 31(4):1418-1459; On inside debt as governance: Li, F., Lin, S., Sun, S., Tucker, A. 2018. Risk-Adjusted Inside Debt. Global Finance Journal 35: 12-42. Or compensation incentives: Core, J. and Guay W., 1999, The use of equity grants to manage optimal equity incentive levels, Journal of Accounting and Economics 28, 151-184. You need to discuss those aspects of possible channels to give readers a more comprehensive view and a richer story and/or point out future research direction from these perspectives.

Response:

Thank you for this real critical comment. We do acknowledge the lack of clarity in the previous manuscript. Following your suggestion, we picked and added the corporate governance perspective into our paper. To be honest, it is impossible for us to integrate the corporate governance into the whole paper within 10 days. Therefore, after carefully discussed our co-authors, we decide to address your comments in our introduction and theoretical implication section. For the sake of brevity, we do not cite the text here. Please see the content from line 42-50 and line 299-313 for details.

R3-4 There are many typos and grammatical mistakes throughout the paper, making it hard to read and understand. For example, in the abstract, “accounting effects strategic alliances” misses “of”. Try to avoid long sentences and vague words. Use short, precise, and concise sentences and be more straightforward.

Response:

Thank you for pointing out the typos in the previous manuscript. We have corrected them in the revision. Again, we have invited a professional copy editor to proofread the manuscript to ensure the quality of the writing. 

R3-5 The last section should be called conclusion where you should summarize all your findings, their implications to researchers and practitioners, future direction for research, limitation of the current study, etc. You need to seriously proofread the paper and extend and update your references.

R3-6 Related to the above point, the paper is minimally developed and too short. You should extend it significantly based on reviewers’ comments.

Response:

Thank you for the constructive comment. Following your suggestions, we renamed the conclusion section and added the theoretical implication. For the sake of brevity, we do not cite the text here. Please see the content from line 299-313 for details.

In conclusion, I would like to thank the authors for a very interesting, unique and potentially important paper. Hope these comments and suggestions can help further their study.

We greatly appreciate your constructive and helpful comments; they encouraged us to think through the theoretical and presentation issues more clearly and better position our core contributions. We hope this substantial revision, in line with your suggestions, sufficiently addresses your concerns. Thank you again for the time and effort you clearly put in to help us improve our paper.

Round 2

Reviewer 1 Report

This study tests the differential impacts of the different types of strategic alliances on firm performance with the event-study method. However, this research method collects only market reaction upon the announcement of strategic alliances and which is readily measurable. Whether the efficiency of the alliance is improving, is still unknown. Unless evidence of efficiency improvement is provided, the conclusion is difficult to determine.

Author Response

Vertical vs. Horizontal: How Strategic Alliance Type Influence Firm Performance?

Sustainability-604157: Response to Comments of Reviewer 1

We provide responses to each of your specific comments (your comments are excerpted in bold fond, followed by our responses in normal type).

R1-1 This study tests the differential impacts of the different types of strategic alliances on firm performance with the event-study method. However, this research method collects only market reaction upon the announcement of strategic alliances and which is readily measurable. Whether the efficiency of the alliance is improving, is still unknown. Unless evidence of efficiency improvement is provided, the conclusion is difficult to determine.

Response:

Thank you for your helpful suggestion. Following your suggestion, we regressed a firm’s operating efficiency, which is estimated as the firm’s operating income/loss over total firm assets, in the first, second, and third year after the announcement year, on firm level control variables. The results are presented in Table 5. For the sake of brevity, we do not cite the text here. Please see the content from line 297-308 for details. Thanks.

Reviewer 3 Report

I’ll give you more time to incorporate my previous comments.

Author Response

Vertical vs. Horizontal: How Strategic Alliance Type Influence Firm Performance?

Sustainability-604157: Response to Comments of Reviewer 3

We provide responses to each of your specific comments (your comments are excerpted in bold fond, followed by our responses in normal type).

R3-1 You should clarify the contributions of the paper which are not elaborated well in the current paper. You can talk about the following contributions: What insights can you provide based on your finding? Do they push forward our understanding? What should we do with your research? Do you have any suggestions to improve the current regulation or practice? Adding the above discussion and extend your literature review may help you make more contributions and position your contributions better.

Response:

Thank you for the constructive comment. In this revision, we made the major changes in our introduction to address your concern. For the sake of brevity, we do not cite the text here. Please see the content from line 42-50 for details.

R3-2 You should study and rationalize the use of firm size measures in the literature since frim size is the key variable in this area. See Dang et al. 2018. Measuring Firm Size in Empirical Corporate Finance. Journal of Banking & Finance, 86:159-176. After all it is the most significant variable in most studies alike. You need to discuss and justify your firm size measure.

Response:

Thanks for your valuable comment. We do acknowledge that the firm size measure is the key variable in this area. Actually, one of the type of strategic alliance is size scope (asymmetric vs. symmetric alliances). Following the previous literature (Kalaignanam, Shankar and Varadarajan, 2007), we consider asymmetric alliances as those alliances in which the ratio of the larger firm’s assets to that of the smaller firm is greater than five.

R3-3 My main suggestion is that you should tell a richer story and link to more literature by discussing more relevant channels. One main channel is corporate governance. The abnormal returns of different alliances may be driven by agency problem, for instance. You should consider, for example, market competition as a governance mechanism: Giroud, X., and H., Mueller, 2011, Corporate governance, product market competition, and equity prices. Journal of Finance 66, 563-600. The interactions between the executives, such as mutual monitoring among the executives: Li, Z.F., 2014, Mutual monitoring and corporate governance, Journal of Banking & Finance, 45, 255-269; Li, Z.F., 2018, Mutual monitoring and agency problem. 
https://www.researchgate.net/publication/272305464_Mutual_Monitoring_and_Agency_Problems; and external interactions between CEOs in the industry tournament: Coles et al. 2018, Industry Tournament Incentives, Review of Financial Studies, 31(4):1418-1459; On inside debt as governance: Li, F., Lin, S., Sun, S., Tucker, A. 2018. Risk-Adjusted Inside Debt. Global Finance Journal 35: 12-42. Or compensation incentives: Core, J. and Guay W., 1999, The use of equity grants to manage optimal equity incentive levels, Journal of Accounting and Economics 28, 151-184. You need to discuss those aspects of possible channels to give readers a more comprehensive view and a richer story and/or point out future research direction from these perspectives.

Response:

Thank you for this real critical comment. We do acknowledge the lack of clarity in the previous manuscript. Following your suggestion, we picked and added the corporate governance perspective, especially resource dependence theory into our paper. To address your comments, in the introduction, we added, “Our fresh findings may extend strategic alliance and corporate governance literature by advancing our understanding of being good governance with business partners to maximize the value of the strategic alliance. Moreover, this study may offer essential suggestions to managers on where to allocate their precious resources and efforts when and how they collaborate with their organizational network partners to enhance competitive advantage.

Then, we also included and integrated the corporate governance perspective into the theoretical foundations and hypothesis section. “….according to the sub-theory of RBV, resource dependence theory (RDT) stated that inter-organizational relationships could also help an organization to reduce environment uncertainty and gain mutual benefits, which used for explaining why firms engage in long-term relationships with other firms…….”

“According to the resource dependence theory (RDT), interdependence is a phenomenon that “exists whenever one actor does not entirely control all of the conditions necessary for the achievement of an action or for obtaining the outcome desired from the action”. As RDT suggests, organizations form inter-organizational relationships with other organizations as a governance mechanism to reduce uncertainty and manage dependence”

Finally, we added the new theoretical implications section:

“This study contributes to the literature in several notable ways. First, this study extends the resource dependence theory (RDT) theory and strategic alliance literature by empirical testing the differential impacts of the firm’s different types of strategic alliance on firm alliance performance. We hope that the idea of integrating different types of strategic alliance may pave the way for future empirical studies to maintain their inter-organizational relationships to gain competitive advantages.

Moreover, this study enriches the research on the marketing-finance interface by investigating that firms’ vertical symmetry alliances gain more abnormal returns than others. Such findings respond Yasuda and Iijima ‘s (2005) call for more research on their matrix in bridging firms’ inter-organizational resources and firm alliance performance.

Third, according to previous corporate governance literature, weak governance firms have lower equity returns, worse operating performance and the lower firm value. The current findings in this study may enrich corporate governance literature by creating and maintaining good governance to maximize the value from their vertical symmetry strategic alliance.”

R3-4 There are many typos and grammatical mistakes throughout the paper, making it hard to read and understand. For example, in the abstract, “accounting effects strategic alliances” misses “of”. Try to avoid long sentences and vague words. Use short, precise, and concise sentences and be more straightforward.

Response:

Thank you for pointing out the typos in the previous manuscript. We have corrected them in the revision. Again, we have invited a professional copy editor to proofread the manuscript to ensure the quality of the writing. 

R3-5 The last section should be called conclusion where you should summarize all your findings, their implications to researchers and practitioners, future direction for research, limitation of the current study, etc. You need to seriously proofread the paper and extend and update your references.

R3-6 Related to the above point, the paper is minimally developed and too short. You should extend it significantly based on reviewers’ comments.

Response:

Thank you for the constructive comment. Following your suggestions, we renamed the conclusion section and added the theoretical implication. For the sake of brevity, we do not cite the text here. Please see the content from line 320-334 for details.

In conclusion, I would like to thank the authors for a very interesting, unique and potentially important paper. Hope these comments and suggestions can help further their study.

We greatly appreciate your constructive and helpful comments; they encouraged us to think through the theoretical and presentation issues more clearly and better position our core contributions. We hope this substantial revision, in line with your suggestions, sufficiently addresses your concerns. Thank you again for the time and effort you clearly put in to help us improve our paper.

Round 3

Reviewer 1 Report

The differential impacts of the different types of strategic alliances on firms' long term performance after the announcement have been tested, and the evidence shows that vertical and symmetry alliances improve a company’s performance significantly and consistently. Therefore, the conclusion is fully supported.

Author Response

Thanks for your positive comments. And appreciate your help.

Reviewer 3 Report

Revision is minimal and not satisfactory.

Author Response

Vertical vs. Horizontal: How Strategic Alliance Type Influence Firm Performance?

Sustainability-604157: Response to Comments of Reviewer 3

We provide responses to each of your specific comments (your comments are excerpted in bold fond, followed by our responses in normal type).

R3-1 You should clarify the contributions of the paper which are not elaborated well in the current paper. You can talk about the following contributions: What insights can you provide based on your finding? Do they push forward our understanding? What should we do with your research? Do you have any suggestions to improve the current regulation or practice? Adding the above discussion and extend your literature review may help you make more contributions and position your contributions better.

Response:

Thank you for the constructive comment. In this revision, we made the major changes in our introduction and add the theoretical implication section to address your concern. In our introduction part, we emphasized, “This study contributes to strategic alliances literature and practice. First, while interdependencies between business partners have been studied in the strategic alliance literature, relationship between alliances with different types have not received due attention in prior studies. The most novel theoretical contribution for this study is that it not only shows the differential impacts of firm’s different types of strategic alliance (industry scope: vertical vs. horizontal alliances and size scope: asymmetric vs. symmetric alliances) on firm alliance performance, but is also discovers firm’s vertical symmetry alliances gain more abnormal returns than others. Our fresh findings may extend strategic alliance and corporate governance literature by advancing our understanding of being good governance with business partners to maximize the value of the strategic alliance. Clearly, combining the different dimensions of alliance portfolios suggest a decent avenue for future research. Moreover, this study may offer essential suggestions to managers on where to allocate their precious resources and efforts when and how they collaborate with their organizational network partners to enhance competitive advantage.”

And in our theoretical implication section, we added, “This study contributes to the literature in several notable ways. First, this study extends the resource dependence theory (RDT) theory [21, 22] and strategic alliance literature [12, 25, 26] by empirical testing the differential impacts of the firm’s different types of strategic alliance on firm alliance performance. We hope that the idea of integrating different types of strategic alliance may pave the way for future empirical studies to maintain their inter-organizational relationships to gain competitive advantages.

Moreover, this study enriches the research on the marketing-finance interface by investigating that firms’ vertical symmetry alliances gain more abnormal returns than others. Such findings respond Yasuda and Iijima ‘s (2005) [11] call for more research on their matrix in bridging firms’ inter-organizational resources and firm alliance performance.

Third, according to previous corporate governance literature [58], weak governance firms have lower equity returns, worse operating performance and the lower firm value. The current findings in this study may enrich corporate governance literature by creating and maintaining good governance to maximize the value from their vertical symmetry strategic alliance.

R3-2 You should study and rationalize the use of firm size measures in the literature since frim size is the key variable in this area. See Dang et al. 2018. Measuring Firm Size in Empirical Corporate Finance. Journal of Banking & Finance, 86:159-176. After all it is the most significant variable in most studies alike. You need to discuss and justify your firm size measure.

Response:

Thanks for your valuable comment. We do acknowledge that the firm size measure is the key variable in this area. Actually, one of the type of strategic alliance is size scope (asymmetric vs. symmetric alliances). Following the previous literature (Kalaignanam, Shankar and Varadarajan, 2007), we consider asymmetric alliances as those alliances in which the ratio of the larger firm’s assets to that of the smaller firm is greater than five.

R3-3 My main suggestion is that you should tell a richer story and link to more literature by discussing more relevant channels. One main channel is corporate governance. The abnormal returns of different alliances may be driven by agency problem, for instance. You should consider, for example, market competition as a governance mechanism: Giroud, X., and H., Mueller, 2011, Corporate governance, product market competition, and equity prices. Journal of Finance 66, 563-600. The interactions between the executives, such as mutual monitoring among the executives: Li, Z.F., 2014, Mutual monitoring and corporate governance, Journal of Banking & Finance, 45, 255-269; Li, Z.F., 2018, Mutual monitoring and agency problem. 
https://www.researchgate.net/publication/272305464_Mutual_Monitoring_and_Agency_Problems; and external interactions between CEOs in the industry tournament: Coles et al. 2018, Industry Tournament Incentives, Review of Financial Studies, 31(4):1418-1459; On inside debt as governance: Li, F., Lin, S., Sun, S., Tucker, A. 2018. Risk-Adjusted Inside Debt. Global Finance Journal 35: 12-42. Or compensation incentives: Core, J. and Guay W., 1999, The use of equity grants to manage optimal equity incentive levels, Journal of Accounting and Economics 28, 151-184. You need to discuss those aspects of possible channels to give readers a more comprehensive view and a richer story and/or point out future research direction from these perspectives.

Response:

Thank you for this real critical comment. We do acknowledge the lack of clarity in the previous manuscript. Following your suggestion, we picked and added the corporate governance perspective, especially resource dependence theory into our paper. To address your comments, in the introduction, we added, “Our fresh findings may extend strategic alliance and corporate governance literature by advancing our understanding of being good governance with business partners to maximize the value of the strategic alliance. Moreover, this study may offer essential suggestions to managers on where to allocate their precious resources and efforts when and how they collaborate with their organizational network partners to enhance competitive advantage.

Then, we also included and integrated the corporate governance perspective into the theoretical foundations and hypothesis section. “….according to the sub-theory of RBV, resource dependence theory (RDT) stated that inter-organizational relationships could also help an organization to reduce environment uncertainty and gain mutual benefits, which used for explaining why firms engage in long-term relationships with other firms…….”

“According to the resource dependence theory (RDT), interdependence is a phenomenon that “exists whenever one actor does not entirely control all of the conditions necessary for the achievement of an action or for obtaining the outcome desired from the action”. As RDT suggests, organizations form inter-organizational relationships with other organizations as a governance mechanism to reduce uncertainty and manage dependence”

Finally, we added the new theoretical implications section:

“This study contributes to the literature in several notable ways. First, this study extends the resource dependence theory (RDT) theory and strategic alliance literature by empirical testing the differential impacts of the firm’s different types of strategic alliance on firm alliance performance. We hope that the idea of integrating different types of strategic alliance may pave the way for future empirical studies to maintain their inter-organizational relationships to gain competitive advantages.

Moreover, this study enriches the research on the marketing-finance interface by investigating that firms’ vertical symmetry alliances gain more abnormal returns than others. Such findings respond Yasuda and Iijima ‘s (2005) call for more research on their matrix in bridging firms’ inter-organizational resources and firm alliance performance.

Third, according to previous corporate governance literature, weak governance firms have lower equity returns, worse operating performance and the lower firm value. The current findings in this study may enrich corporate governance literature by creating and maintaining good governance to maximize the value from their vertical symmetry strategic alliance.”

R3-4 There are many typos and grammatical mistakes throughout the paper, making it hard to read and understand. For example, in the abstract, “accounting effects strategic alliances” misses “of”. Try to avoid long sentences and vague words. Use short, precise, and concise sentences and be more straightforward.

Response:

Thank you for pointing out the typos in the previous manuscript. We have corrected them in the revision. Again, we have invited a professional copy editor to proofread the manuscript to ensure the quality of the writing. 

R3-5 The last section should be called conclusion where you should summarize all your findings, their implications to researchers and practitioners, future direction for research, limitation of the current study, etc. You need to seriously proofread the paper and extend and update your references.

R3-6 Related to the above point, the paper is minimally developed and too short. You should extend it significantly based on reviewers’ comments.

Response:

Thank you for the constructive comment. Following your suggestions, we renamed the conclusion section and added the theoretical implication.

We also followed the editor’s suggestion and added some more statement in the literature review section. Page 3, we added, The similarity of these two types of alliances lies in the following several perspectives: (1) partners are confident with the advice and recommendations suggested by trusted business relationships [11, 30, 59], so such organizational relationship may reduce uncertainty and costs in business transactions; (2) information sharing with business partners can foster knowledge spillover and production [30]; (3) by involving relationship participators to co-work through problems, joint problem-solving arrangements may replace the simple exit-or-stay response of the market players [49, 60]. While, these two types of alliances also have distinct features: (1) key partners, vertical alliances mainly focus on their suppliers and customers, while horizontal alliances lies on their main competitors; (2) key activities, vertical alliances try to co-operate with their partners by sharing raw materials, production, and distribution to maximize profits, while in case of horizontal alliances, by reducing vicious competition, common potential market opportunity is pursued to create synergies among competitors. (3) Key perspectives, high quality of vertical alliances can offer complementary information and knowledge to achieve organizational cooperation; although the information and knowledge in the same industry are similar, firms in many knowledge intensive industries force themselves to seek strategic alliances with major competitors with whom they have joint interests in some markets and/or product fields [38, 60]

Page 4, we added,Previous research suggests that size of a potential partner is an important criterion in partner selection. Firstly, let’s take a look at the motivation of symmetric alliances. A possible reason was that both firms would place the same importance on the alliance, and bargaining power would be almost equal. Similarly, Williams and Lilley (1993) argued strongly that alliance may have the best chance of long-term success when both partners are comparable in sophistication and size [40]. While, on the other hand, more previous research analyzed the asymmetric distribution of common and private benefits in alliances has underscored the incentives that such benefits provide for continued collaboration. The major reason for asymmetric alliances is to access complementary resources from each other [41]. For example, small biotech firms often form alliances with large pharmaceutical firms with the purpose of utilizing the latter’s expertise in the U.S. Food and Drug Administration agency’s approval process and in market coverage.

In our limitation and future research, we also addedThis study has several limitations. First, by choosing Yasuda and Iijima (2005)’s alliance matrix as the foundation of this study, we only focus on these two main types of strategic alliances as the search terms in the present study and some other matrix may be inevitably excluded. In the future, we want to create a broader and complex matrix to help us to identify important contributions using other types of strategic alliances. Moreover, we obtain data from Bloomberg M&A database through 2010/1/1 to 1/1/2016 and get only 305 alliances announcements of joint ventures. The limited sample might have reduced the statistical power necessary to generate more significant findings. Further research could test our hypotheses using larger samples. Third, this study mainly focused to explore the relationship between relationship between different types alliances and firm performance. More and more empirical tests may also provide insights on the firm-level antecedents of different types alliances. Finally, it is only focused on the bright side of strategic alliances. While there are lots of failure cases about strategic alliances or some strategic alliances duration is very short. In the future, an attempt to find some boundary conditions and empirical tests will be interested in the marketing or management discipline.

In conclusion, I would like to thank the authors for a very interesting, unique and potentially important paper. Hope these comments and suggestions can help further their study.

We greatly appreciate your constructive and helpful comments; they encouraged us to think through the theoretical and presentation issues more clearly and better position our core contributions. We hope this substantial revision, in line with your suggestions, sufficiently addresses your concerns. Thank you again for the time and effort you clearly put in to help us improve our paper.
